

# A framework for expanding aqueous chemistry in the Community Multiscale Air Quality (CMAQ) model version 5.1

Kathleen M. Fahey[1], Annmarie G. Carlton[2], Havala O.T. Pye[1], Jaemeen Baek[3], William T. Hutzell[1], Charles Stanier[4], Kirk R. Baker[5], K. Wyat Appel[1], Mohammed Jaoui[6], John H. Offenberg[6]

[1]Computational Exposure Division, National Exposure Research Laboratory, Office of Research and Development, U.S. Environmental Protection Agency, Research Triangle Park, North Carolina
[2]Department of Chemistry, University of California, Irvine, Irvine, CA
[3]Formerly at the Department of Chemical and Biochemical Engineering, University of Iowa, Iowa City, IA
[4]Department of Chemical and Biochemical Engineering, University of Iowa, Iowa City, IA
[5]Air Quality Assessment Division, Office of Air Quality Planning and Standards, Office of Air and Radiation, U.S. Environmental Protection Agency, Research Triangle Park, North Carolina
[6]Exposure Methods and Measurements Division, National Exposure Research Laboratory, Office of Research and Development, U.S. Environmental Protection Agency, Research Triangle Park, North Carolina

*Correspondence to*: Kathleen M. Fahey (fahey.kathleen@epa.gov)

**Abstract.** This paper describes the development and implementation of an extendable aqueous phase chemistry option (AQCHEM-KMT(I)) for the Community Multiscale Air Quality (CMAQ) modeling system, version 5.1. Here the Kinetic PreProcessor (KPP), version 2.2.3, is used to generate a Rosenbrock solver (Rodas3) to integrate the stiff system of ODEs that describe the mass transfer, chemical kinetics, and scavenging processes of CMAQ clouds. CMAQ's standard cloud chemistry module (AQCHEM) is structurally limited to the treatment of a simple chemical mechanism. This work advances our ability to test and implement more sophisticated aqueous chemical mechanisms in CMAQ and further investigate the impacts of microphysical parameters on cloud chemistry.

Box model cloud chemistry simulations were performed to choose efficient solver and tolerance settings, evaluate the implementation of the KPP solver, and assess the direct impacts of alternative solver and kinetic mass transfer on predicted concentrations for a range of scenarios. Month-long CMAQ simulations for winter and summer periods over the U.S. reveal the changes in model predictions due to these cloud module updates within the full chemical transport model. While monthly average CMAQ predictions are not drastically altered between AQCHEM and AQCHEM-KMT, hourly concentration differences can be significant. With added in-cloud secondary organic aerosol (SOA) formation from biogenic epoxides (AQCHEM-KMTI), normalized mean error and bias statistics are slightly improved for 2-methyltetrols and 2-methylglyceric acid at the Research Triangle Park measurement site in North Carolina during the SOAS field campaign period. The added in-cloud chemistry leads to a monthly-average increase of 11-18% in "cloud" SOA at the surface in the eastern U.S. for June 2013.



## 1 Introduction

Clouds and fogs impact the amount, composition, and spatial distribution of trace atmospheric species through a complex interplay of chemistry and physics. Pollutants are transported via convection and wet deposition (Barth et al., 2001; Wonaschuetz et al., 2012; Yin et al., 2001) and altered through condensed phase chemistry (Graedel and Weschler, 1981;

Lelieveld and Crutzen, 1991). Water droplets offer a medium for soluble gases to dissolve, dissociate, and undergo aqueous phase chemical reactions. This is well established for the conversion of gas phase $SO_2$ to particle phase sulfate ($SO_4^{2-}$) (Martin, 1984; Martin and Good, 1991). Atmospheric $SO_4^{2-}$ is an important component of fine aerosol mass and is a known contributor to adverse effects on human health and ecosystems. In an environment where clouds or fogs are present, aqueous phase production of $SO_4^{2-}$ dominates over production in the gas phase (Seigneur and Saxena, 1988; Ervens, 2015), and decades of

research have been devoted to studying the impacts of aqueous production of acidic species, like $SO_4^{2-}$, on acid deposition, including effectively representing that aqueous production in models (Chang et al., 1987; Walcek and Taylor, 1986; Pandis and Seinfeld, 1989; Fahey and Pandis, 2001; Gong et al., 2011; Giulianelli et al., 2014; Herckes et al., 2015). More recent studies have focused on the potentially significant role that aqueous pathways may have on the formation of secondary organic aerosol (SOA) (McNeill et al., 2012; Ervens et al., 2014; Ervens et al., 2011; Lim et al., 2005; Liu et al., 2012; Carlton et

al.,2008). It has been proposed that cloud water provides a medium for the production of highly oxidized organic compounds that remain in the aerosol phase after cloud droplet evaporation, contributing to secondary organic aerosol mass (Ervens et al., 2011; McNeill, 2015; Ervens, 2015). Sulfate and organic components can contribute more than half of the total $PM_{2.5}$ concentration in many regions across the globe (Philip et al., 2014; Jimenez et al., 2009; Hansen et al., 2003; Brewer and Adlhoch, 2005).

The degree to which species enter the aqueous phase and participate in cloud processing is dependent upon species' intrinsic chemical properties, such as solubility and reactivity, and also upon microphysical characteristics of the droplet spectrum (Schwartz, 1986; Schwartz and Frieberg, 1981; Sander, 1999; Pandis et al., 1990; Fahey et al., 2005). The processes of droplet activation, scavenging, and chemical production lead to shifts in the aerosol size/composition distribution after droplet

evaporation. These changes to the aerosol distribution then further impact aerosol transport and microphysics along with the direct and indirect radiative effects associated with that aerosol (Kreidenweis et al., 2003). The chemistry and (micro)physics of clouds and fogs must therefore be well represented in models in order to effectively assess the impacts of emissions changes on future air quality and climate.

The Community Multiscale Air Quality (CMAQ) modeling system (Byun and Schere, 2006) is a widely-used state-of-the-science chemical transport model, applied on a range of scales for research, regulatory, and forecasting purposes. In the United States, it is among the most commonly used air quality models in attainment demonstrations for National Ambient Air Quality Standards for ozone and $PM_{2.5}$ (US EPA, 2007). In such frameworks, air quality simulations may be required for many



emissions scenarios, large domains, and/or long time periods. It is important therefore that the selected modeling system be as efficient as possible while also being able to faithfully capture the most important chemical and physical processes affecting the pollutants of interest and their response to emissions changes.

Due in part to computational constraints, historically only a simple description of aqueous phase chemistry has been implemented in many regional air quality models. Cloud chemistry in CMAQ (AQCHEM), for example, is based on the cloud module of the Regional Acid Deposition Model (RADM) (Walcek and Taylor, 1986) with minimal updates to the mechanism in recent years (Carlton et al., 2008). When the cloud chemistry module is called, species are distributed between gas, interstitial aerosol, and aqueous phases instantaneously depending upon the initial modal aerosol distribution and solubility.

At each time step, a bisection method is used to solve for the bulk droplet pH and the associated phase/ionic distribution of model species based on known total concentrations and assuming electroneutrality and thermodynamic equilibrium. Activity coefficients, estimated with the Davies equation, are applied to ionic species in solution. A forward Euler method is used to solve a set of seven oxidation reactions, with time stepping based on the reaction rates and precursor/oxidant concentrations for S(VI) production alone. Scavenging of interstitial aerosol and wet deposition is calculated alongside the chemical kinetics

and mass transfer (Binkowski and Roselle, 2003). Because the mechanism is hard-coded into the solver, as well as the solver's potential stability issues when applied to stiff systems of ODEs, it is difficult to expand CMAQ's current cloud chemistry treatment to additional complex chemistry for other species. It has long been understood, however, that the aqueous phase chemistry of clouds and aerosols affects a myriad of species, and we may be insufficiently representing cloud chemistry with our simple mechanism geared mainly towards sulfur oxidation. While computational efficiency remains important, it is also

crucial for models to represent important new scientific discoveries and newly understood physicochemical processes faithfully. As computational capabilities expand and field and laboratory studies continue to elucidate additional potentially important atmospheric aqueous chemistry pathways, it is increasingly important to maintain a modeling framework that allows for ready expansion to and investigation of that chemistry.

With these motivations, the Kinetic PreProcessor (KPP) version 2.2.3 (Damian et al., 2002) has been applied to generate a Rosenbrock solver for the CMAQ cloud chemistry mechanism (AQCHEM-KMT) as well as an expanded mechanism that includes additional aqueous secondary organic aerosol formation from biogenic-derived epoxides (Pye et al., 2013) in cloud (AQCHEM-KMTI). The KPP implementation includes kinetic mass transfer between the gas and aqueous phases, dissociation/association, chemical kinetics, interstitial aerosol scavenging, and wet deposition and is readily expandable to

larger chemical mechanisms. In the following sections, the details of the development and implementation of these additional aqueous-phase chemistry options are presented. In section 2, the AQCHEM-KMT/AQCHEM-KMTI structure is detailed alongside a description of the box model testing used in choosing solver parameters and examining the direct impacts of kinetic mass transfer and alternate solvers on predicted concentrations. In section 3, the impacts of the updated aqueous chemistry options are examined for winter- and summer- month regional CMAQ simulations. Some of the benefits and drawbacks of



these newly implemented cloud chemistry options are discussed in section 4, along with some directions for future development work and applications.

**2 Aqueous-phase chemistry model description and box model testing**

Table 1 contrasts the main features of the three cloud chemistry options examined here. Building upon the approach outlined
by Baek et al. (2011), we used the Kinetic PreProcessor version 2.2.3 (KPP) to automatically generate Fortran90 code for the numerical integration of the CMAQ aqueous phase chemical mechanism (Damian et al., 2002; Sandu and Sander, 2006). KPP is a free software tool that translates chemical mechanism information (e.g., species, reactions, rate coefficients) into Fortran90, Fortran77, Matlab, or C code to efficiently integrate chemical kinetics. KPP includes multiple stiff numerical integrators and has a modularity that allows rapid and straightforward testing of alternative solvers and mechanisms. It may also be used to
generate the tangent linear or adjoint models for a given system, but this capability is not investigated here. Minor changes were made to the generated code to account for our system and I/O requirements. The model driver (for 0D box model and CMAQ implementation) was developed outside of KPP.

AQCHEM-KMT solves the processes of phase transfer, chemical kinetics, ionic dissociation/association, scavenging of
interstitial aerosol, and wet deposition. Here AQCHEM-KMT maintains the same initialization and post-cloud redistribution assumptions as in AQCHEM, including (1) at the start of cloud processing, accumulation and coarse mode aerosols are instantaneously activated to droplets, all $N_2O_5(g)$ is converted to $HNO_3(g)$, and all $H_2SO_4(g)$ is transferred to aqueous phase S(VI); (2) at the end of cloud processing, $HNO_3(g)$ and accumulation mode aerosol $NO_3^-$, as well as $NH_3(g)$ and accumulation mode aerosol $NH_4^+$, are redistributed to retain their initial (i.e., pre-cloud) gas/aerosol phase distributions; and (3) all non-
volatile aqueous phase mass production is added to the accumulation mode (Binkowski and Roselle, 2003). Initial gas and aqueous concentrations (in units of molecules/cm$^3$ air) are calculated based on their initial phase distribution, temperature/pressure, and any additional simplifying assumptions that may apply (e.g., $H_2SO_4$ mentioned above). As in AQCHEM, the hydroxyl radical (OH) concentration is kept constant, with droplet concentrations estimated from the initial gas phase OH using Henry's Law. Initial [$H^+$] is estimated from an ion balance on the instantaneously activated ionic species,
$OH^-$, and completely dissolved gaseous species (i.e., $H_2SO_4(g)$) at t = 0 seconds. $H^+$ and other gas, aqueous, and aerosol species' concentrations then evolve dynamically for the duration of cloud processing.

After initialization, AQCHEM-KMT solves the following system of ODEs:

$$\frac{dC_{g,i}}{dt} = -k_{mt,i}w_L C_{g,i} + \frac{k_{mt,i}}{H_i RT} C_{aq,i} \qquad (1)$$

$$\frac{dC_{aq,i}}{dt} = k_{mt,i}w_L C_{g,i} - \frac{k_{mt,i}}{H_i RT} C_{aq,i} + P_{aq,i} - L_{aq,i} C_{aq,i} \qquad (2)$$





$$\frac{dC_{aaero,i}}{dt} = -L_{aaero,i}C_{aaero,i}$$

(3)

where

$C_{g,i}$ = gas phase concentration of species i (molecules/cm$^3$ air)

$C_{aq,i}$ = aqueous phase concentration of species i (molecules/cm$^3$ air)

$C_{aaero,i}$ = interstitial (Aitken) aerosol concentration of species i (molecules/cm$^3$ air)

$k_{mt,i}$ = mass transfer coefficent of species i

$w_L$ = liquid water content fraction (cm$^3$ H$_2$O / cm$^3$ air)

$H_i$ = Henry's Law coefficient of species i

R = gas constant

T = temperature (K)

$P_{aq,i}$ = rate of production of species i in the aqueous phase. This includes contributions from chemical reactions ($R_{aq}$), scavenging of interstitial aerosol ($A_{scav}$), and dissociation/association of ionic species ($X_{ion,f/b}$)

$L_{aq,i}$ = loss term for aqueous species i. This includes loss due to chemical reactions ($R_{aq}$), dissociation/association of ionic species ($X_{ion,f/b}$), and wet deposition ($W_{dep}$)

$L_{aaero,i}$ = loss term for interstitial (Aitken) aerosol species i due to scavenging by cloud droplets ($A_{scav}$)

The rate expressions for these processes are further detailed in Table 2. The first two terms in equations 1 and 2 represent the concentration changes due to mass transfer between the gas and aqueous phases. While in AQCHEM, Henry's Law equilibrium is assumed to be instantaneously reached for all species, species distributions between the phases may deviate

significantly from equilibrium (Audiffren et al., 1998; Sander, 1999; Gong et al., 2011). As in Schwartz (1986), the combined impacts of gas-phase diffusion and interfacial mass transport limitations are incorporated into a single mass transfer coefficient ($k_{mt}$), the expression for which is given in Table 2. Table S1 lists the mass transfer "reactions" considered here, as well as the constants used in the mass transfer coefficient calculation.

Once in the droplet, species are allowed to dissociate into ions, represented here as forward and reverse reactions, as well as participate in irreversible chemical reactions. The hydrogen ion, [H+], crucial in determining species' phase and ionic distributions and reaction rates, is allowed to evolve dynamically from its initial value. Ionization and chemical kinetic reactions and associated rate coefficients are listed in tables S2 and S3 respectively. Concentration gradients may develop



within the droplet for some species that participate in rapid aqueous phase reactions (e.g., $O_3$ (Jacob, 1986; Walcek and Taylor, 1986)). In such cases a correction factor Q may be applied to account for aqueous phase diffusion limitations on the overall reaction rate (Table 2) (Schwartz and Freiberg, 1981) This factor is also applied to other species to maintain consistent treatment between aqueous aerosol and cloud droplet chemistry (i.e., IEPOX/MAE (Pye et al., 2013)). Box model tests

indicate, however, that aqueous diffusion impacts on the evolution of the chemical system are minimal. As the chemical mechanism evolves, aqueous diffusion limitations may become more important and will be revisited. Wet deposition (of all aqueous species) and interstitial aerosol scavenging (for Aitken mode aerosol species) are represented as first-order loss processes. Additional information on their rate coefficients can also be found in Table 2. A list of CMAQ and "local" AQCHEM-KMT(I) species is given in Table S4.

## 2.1 Solver selection and tolerance settings

A previous comparison of several stiff ODE solvers applied for different types of atmospheric chemical systems indicated that Rosenbrock solvers are some of the most efficient at solving computationally-intensive multiphase chemistry problems for modest accuracies (Sandu et al., 1997 a,b). KPP offers several Rosenbrock integrator options. In an effort to determine the optimal solver for our mechanism, we applied each KPP Rosenbrock solver with a positive definite adjustment (Sander et al.,

2011) to over 20000 scenarios (Table S5) representing a range of atmospheric conditions and then compared the results to a reference solution generated with the Variable-coefficient Ordinary Differential Equation solver (VODE). VODE is an initial-value ODE solver that uses variable-coefficient Backward Differentiation Formula (BDF) methods for stiff systems and may be viewed as a successor to LSODE (Hindmarsh, 1983), which historically has been commonly applied to generate reference solutions for atmospheric chemistry problems (Brown et al., 1989; Sandu et al., 1997b). DVODE (VODE with double

precision) was downloaded from the netlib repository (http://www.netlib.org) and applied to the rate equations and Jacobian for the system of AQCHEM-KMT reactions to generate a reference solution as well as provide an independent check on the codes generated with KPP. The reference solution for each scenario was generated using relative and absolute tolerance settings of $10^{-8}$. For a subset of the scenarios, the reference was compared with results using the order five Runge Kutta solver, RADAU5 (relative and absolute tolerances set at $10^{-8}$ and $10^{-4}$ molecules/cm$^3$ respectively). RADAU5 is a robust solver used

to provide reference solutions in the previous solver intercomparison studies for atmospheric chemical systems (Sandu et al., 1997a; Sandu et al., 1997b). Available Rosenbrock solver versions, ROS2, ROS3, ROS4, Rodas3, and Rodas4, were applied to the test scenarios for a range of absolute and relative tolerances (absolute tolerance = $10^{-4}$ – $10^4$ molecules/cm$^3$ air, relative tolerances = $10^{-4}$ – $10^{-1}$). As in Sandu et al. (1997a,b), we use significant digits of accuracy (SDA) as a metric to describe the relative accuracy of a solver for a given tolerance set. SDA can be defined as follows:

$$SDA_{min} = -log_{10}\big(max(ER_k)\big) \tag{4}$$

where k represents the CMAQ species involved in aqueous chemistry and



$$ER_k = \sqrt{\frac{1}{N}\sum \left| \frac{C_{i,k,ref}-C_{i,k}}{C_{i,k,ref}} \right|^2} \qquad (5)$$

for N total i scenarios. $C_{i,k,ref}$ is the reference solution from DVODE (or RADAU5) for species k and scenario i, and $C_{i,k}$ is the concentration generated with a Rosenbrock solver for a particular tolerance set. This calculation is limited to concentrations exceeding $10^7$ molecules/cm³ to avoid the influence of large relative errors for very small concentrations.

Figure 1a gives the $SDA_{min}$ for each of the solver/tolerance combinations versus the total CPU time required for all scenarios. This is the value for the single species that deviates furthest from the reference solution. The reference solution here is given by DVODE. Most of the Rosenbrock solvers perform similarly (ROS2 is the exception), with Rodas3 and ROS3 the most efficient at lower accuracies. If an SDA of 2 corresponds to an "accuracy" of 1%, the dashed horizontal line in Figure 1a

represents an accuracy of ~5%. Plots here are shown only for DVODE. For the subset of scenarios tested, DVODE and RADAU5 produce results that are within 0.002% of each other for all scenarios ($SDA_{min} = 5.6$, with no minimum concentration setting).

Rodas3 was implemented as the default integrator in AQCHEM-KMT due to favorable performance compared to the other

Rosenbrock solvers for the scenarios considered here; however, any KPP Rosenbrock solver may be invoked with a change to a single argument in the call to the integrator. Rodas3 "tolerance contours" are given in Figure 1b. While choice of relative tolerance has a significant influence on the accuracy of the results, the solution is comparatively unaffected by the choice of absolute tolerance in the tested range, with degradation in the solution only beginning to occur at absolute tolerances exceeding $10^2$ molecules/cm³ for the relative tolerances tested here. In an effort to be efficient while still maintaining an accuracy well

within 5% of the reference solution, we selected a default absolute tolerance of $10^2$ molecules/cm³ air, a relative tolerance of $10^{-2}$ for most species, and a relative tolerance of $10^{-3}$ for hydrogen peroxide and glyoxal/methylglyoxal. The $SDA_{min}$ for this set of tolerance settings for Rodas3 is represented by the red gridded square in Figure 1. These tolerance settings lead to ~ 2% or better accuracy for all species for the tested scenarios, with most species well under 1%. This selection strikes a balance between accuracy and efficiency for this mechanism. For other applications or in the case of a chemical mechanism expansion,

these values can and should be reevaluated and adjusted as necessary to maintain that balance.

## 2.2 Impacts of mass transfer assumptions and solver on box model predictions

While assuming instantaneous Henry's law equilibrium to describe partitioning between gas and aqueous phases can reduce the often significant computational burden associated with simulating heterogeneous chemistry, past studies have indicated that there are species and conditions for which equilibrium conditions are not met during the lifetime of typical cloud droplets.

In such cases a kinetic mass transfer treatment may be necessary to accurately describe the phase distribution between gas and



cloud or fog droplets and subsequent chemistry (Djouad et al., 2003; Audiffren et al., 1998; Audiffren et al., 1996; Chaumerliac et al., 2000; Ervens et al., 2003). Here we treat mass transfer kinetically as a default in AQCHEM-KMT in an effort to assess how deviations from instantaneous Henry's law equilibrium impact predicted concentrations for short- and long-term averaging periods which may be of interest in different applications.

Box model versions of AQCHEM and AQCHEM-KMT were compared for the Table S5 scenarios to better understand the potential impacts of differing solver and mass transfer treatment on aqueous phase chemistry predictions, isolated from other processes in CMAQ that might complicate the analysis. Here we focus mainly on the predictions for the two species chemically produced in the standard AQCHEM mechanism, $SO_4^{2-}$ and SOA from cloud processing of α-dicarbonyls (ORGC). Figures

2a and 2c show $SO_4^{2-}$ and ORGC predictions for AQCHEM-KMT (assuming a default droplet diameter of 16 μm) versus standard AQCHEM. For many scenarios it appears that the equilibrium assumption is a good one, particularly for $SO_4^{2-}$, with many points falling along or not far from the 1:1 line. There are also significant deviations for several scenarios, especially so for cloud SOA. Figures 2 b and d show the impact of changing the default droplet diameter (to 5 and 30 μm) for $SO_4^{2-}$ and ORGC, respectively. As the droplet diameter increases, deviation between AQCHEM-KMT and AQCHEM predictions for

both $SO_4^{2-}$ and ORGC increases as well. In the case of ORGC, even at small droplet diameters, there can be a large discrepancy between the models. The difference in predicted ORGC concentrations at small droplet diameters is, in part, attributable to the difference between the time-stepping technique of AQCHEM and the Rosenbrock solver in AQCHEM-KMT. AQCHEM steps forward in time based on the rate of $SO_4^{2-}$ production and the lifetime of the cloud. These constraints can produce large time steps that can lead to higher SOA predictions. In AQCHEM-KMT, the solver determines forward time steps based on a

convergence test dependent on all species and their tolerance settings. Different predictions of ORGC also occur for larger cloud droplets when the deviation from equilibrium due to mass transfer limitations may become significant (Figure S1).

The differences in predicted $SO_4^{2-}$ and ORGC with different droplet size (Figures 2 and S1) and $SO_4^{2-}$ with different values for initial pH (Figure S2) (which represents the impact of the activated aerosol fraction) point to the potential importance of

microphysical parameters on predicted concentrations and supports the development of better linkages between aqueous chemistry and cloud microphysics codes. Other species typically show smaller differences for different droplet sizes/pH differences, but that may change with the addition of new chemistry.

### 2.3 Aqueous SOA from biogenic epoxides

Using KPP to generate the code for the updated cloud chemistry module allows for straightforward extension to additional

aqueous chemistry. Here we investigated the expansion of the cloud chemistry mechanism to include the in-cloud formation of SOA from biogenic epoxides (AQCHEM-KMTI). The process is based on reactions in aerosol water incorporated into



CMAQv5.1 (Pye et al., 2013). While acid-catalyzed aqueous SOA formation from species like isoprene epoxydiols (IEPOX) is expected to be more important in highly concentrated aerosol water than comparatively dilute cloud droplets (where oxidation of soluble organic compounds by the hydroxyl radical is expected to be the dominant contributor to SOA mass), some SOA may still be formed in cloud droplets from IEPOX (McNeill, 2015 – Figure 2). Overall the aqueous phase reaction

mechanism in aerosol and cloud water is expected to be similar, but dominant reaction pathways may change between concentrated and dilute conditions (McNeill, 2015). We include these reactions here to improve consistency between CMAQ's cloud and aerosol aqueous chemistry mechanisms and to quantify the potential impacts of adding these cloud water reactions. The additional species and reactions are given in the shaded sections of tables S1, S3, and S4 in the Supplemental Information.

**3 CMAQ simulations and measurements**

The impacts of the updated solver and kinetic mass transfer treatment of AQCHEM-KMT and the additional aqueous chemistry of AQCHEM-KMTI were investigated using multiple CMAQ simulations. Simulation periods and domains were selected based on the availability of model inputs and/or specialized observations for comparison. Winter and summer periods were run to illustrate any seasonal differences in sensitivity to the different cloud chemistry modules. Simulations were conducted for winter and summer 2011 over the contiguous US (CONUS) with inputs developed to evaluate CMAQv5.1 (Appel et al.,

2016), as well as for a summer period over the eastern U.S. coinciding with a 2013 measurement campaign that focused on SOA formation. The CONUS simulations were used to assess the impacts of kinetic mass transfer and solver (AQCHEM vs. AQCHEM-KMT); while the summer simulation over the eastern U.S. was geared towards investigating the impact of additional in-cloud SOA formation from biogenic epoxides (AQCHEM-KMT vs. AQCHEM-KMTI). These CMAQ simulations are further detailed below.

Test 1: CMAQv5.1 was applied with both standard AQCHEM and AQCHEM-KMT cloud chemistry for January and July 2011 over CONUS at 12-km horizontal resolution and 35 vertical layers extending up to 50 mb. Meteorological fields were generated with the Weather Research and Forecasting (WRF) model (Skamarock et al., 2008), version 3.7, and emissions were based on the 2011 National Emissions Inventory (NEI), version 2 (https://www.epa.gov/air-emissions-inventories/2011-

national-emissions-inventory-nei-documentation). Aerosol treatment and gas-phase chemistry were described by the AERO6 aerosol module and CB05e51 carbon bond chemical mechanism respectively (http://www.airqualitymodeling.org/cmaqwiki/index.php?title=CMAQ_v5.1_CB05_updates). Additional simulation configuration options included inline biogenic emissions, inline plume rise, windblown dust, lightning NOx emissions, bidirectional ammonia exchange, and in-line calculation of photolysis rates including absorption and scattering from predicted

gas and aerosol concentrations. More detailed information about these CMAQ options can be found in the CMAQ operational guidance and technical documentation available at www.cmascenter.org. Results are examined after 10 days of spin-up from





clean default initial conditions. Hourly lateral boundary conditions were taken from a global GEOS-Chem simulation from the same period (Henderson et al. 2014).

Test 2: The impact of adding cloud water SOA formation from IEPOX and MPAN products, methacrylic acid epoxide (MAE)
and hydroxymethylmethyl-α-lactone (HMML), was examined for a summer period coinciding with the Southern Oxidant and Aerosol Study (SOAS) (http://soas2013.rutgers.edu/). Two simulations were performed with AQCHEM-KMT and AQCHEM-KMTI for June 1 – July 15, 2013, with 11 days of spin-up. The base model was CMAQv5.0.2+, an interim version of CMAQ between the 5.0.2 and 5.1 official releases. Gas-phase and aerosol chemistry was simulated with the SAPRC07TIC chemical mechanism (Xie et al., 2013, Lin et al., 2013) and AERO6i aerosol module respectively
(http://www.airqualitymodeling.org/cmaqwiki/index.php?title=CMAQ_v5.1_SAPRC07tic_AE6i). The model was applied at a 12-km horizontal resolution over the eastern United States with 35 vertical layers up to 100 mb. Emissions were generated using the 2011 NEI with 2013-specific EGU continuous emission monitoring system (CEMS) data and an offline application of the Biogenic Emission Inventory System (BEIS) version 3.6.1 with BELD4 land cover and vegetation. Wildfire and prescribed fire emissions are based on the Satellite Mapping Automated Reanalysis Tool for Fire Incident Reconciliation
(SMARTFIRE) system (http://www.airfire.org/smartfire/) using 2013 day-specific satellite detection of fires. Windblown dust and lightning NOx emissions were not included, nor was bidirectional ammonia surface exchange. Meteorological inputs were generated with WRFv3.6.1. Lateral boundary conditions and initial conditions were taken from a 36-km resolution CMAQ simulation performed over CONUS, southern Canada, and northern Mexico for the same period. The boundary conditions for that coarser resolution CMAQ simulation were derived from a GEOS-Chem simulation performed at 2°×2.5° lateral resolution.
Additional details on the modeling platform can be found in Pye et al. (2015). Simulated concentrations for 2-methyltetrols and 2-methylglyceric acid (2-MG) (i.e., two SOA products from IEPOX and MAE/HMML) were compared to measurements collected at a site in Research Triangle Park (RTP), NC, June 1 through July 15, 2013. The observed 2-methyltetrol and 2-MG concentrations were obtained via GC-MS analysis of aerosol mass from daily filter measurements using a similar methodology to that described in Lewandowski et al. (2013), Kleindienst et al. (2013), and Edney et al. (2005).

## 3.1 Impact of kinetic mass transfer and Rodas3 solver

Figures 3 and 4 show the difference in predictions between AQCHEM (base) and AQCHEM-KMT ([KMT] – [Base]) for fine $SO_4^{2-}$ and ORGC respectively. The figures include a map of monthly average (a,c) and maximum hourly (b,d) differences for January (top) and July (bottom) 2011. In both winter and summer months, the monthly average difference in $SO_4^{2-}$ concentration is low and does not exceed 0.2 $\mu g/m^3$ for the model periods investigated here. Any impacts of kinetic mass
transfer or solver changes on predicted $SO_4^{2-}$ concentrations for our standard (and relatively simple) cloud chemistry mechanism are diluted when moving from aqueous chemistry box modeling to a regional modeling framework where species are impacted by additional processes and averaging periods are longer. Similarly, there are minimal differences in the winter and summer average ORGC concentrations as well. Hourly concentrations can show more significant differences for both



species, however, with hourly concentration differences as high as 14.7 µg/m$^3$ for SO$_4$$^{2-}$ and 0.4 µg/m$^3$ for ORGC. Similar to clouds themselves, these impacts are rather spatially and temporally variable. To illustrate the potential short-term differences between the base and "KMT" aqueous chemistry modules, Figures 5 a and b show the time series of the SO$_4$$^{2-}$ concentration differences at the grid cell with the highest hourly concentration difference for all hours of the January and July 2011

simulations, respectively. The figures also include modeled total liquid water content values. For most hours, the SO$_4$$^{2-}$ concentrations are very similar for the two runs. However, when the liquid water content values become significant and the cloud chemistry module is called, the hourly differences between predicted SO$_4$$^{2-}$ concentrations can be at least a couple micrograms per cubic meter in magnitude, with kinetic mass transfer leading nearly always here to lower predicted SO$_4$$^{2-}$ concentrations due to the incorporation of gas and interfacial mass transfer limitations. During the rare times when AQCHEM-

KMT predicted higher SO$_4$$^{2-}$ than the base in these cells, the hourly increase in SO$_4$$^{2-}$ was less than 0.26 µg/m$^3$. It should be noted that since CMAQ already tends to have a slightly low bias with respect to SO$_4$$^{2-}$ concentrations in the winter and summer for most regions in the U.S. (Appel et al., 2016), without additional updates to chemistry or improved cloud parameter predictions, AQCHEM-KMT will lead to a small increase in absolute bias for SO$_4$$^{2-}$ at most surface sites in our domain compared to AQCHEM.

ORGC mass predictions are less impacted than SO$_4$$^{2-}$, but this may be due in part to CMAQ's implementation of cloud SOA formation. In AQCHEM-KMT (as in AQCHEM), ORGC is formed from the reaction of glyoxal and/or methylglyoxal with the hydroxyl radical. The hydroxyl radical concentration is estimated at the start of cloud processing based on the initial gas phase concentration (Henry's law) and held constant for the duration of the "master" cloud time step. This was done in part

to compensate for the lack of a more complete treatment of radical/organic chemistry in the aqueous phase, along with a relatively loose coupling between gas and aqueous chemistry in CMAQ. A constant oxidant concentration may cause an artificially high rate of consumption of the precursor species and insensitivity of the reaction to droplet size and associated mass transfer limitations. If a more explicit cloud SOA mechanism was included where OH concentrations were allowed to vary, mass transfer limitations would likely have a greater influence on in-cloud SOA production.

**3.2 Impact of cloud SOA formation pathway from biogenic epoxides**

More than half of fine particulate mass (PM$_{2.5}$) can be made up of organic compounds, with a significant secondary fraction, depending on location and season. While it is an inherently difficult system to fully characterize due to the number of compounds involved, progress has been made in identifying precursor compounds and important pathways leading to SOA formation. Recently SOA formation from isoprene epoxydiols (IEPOX) and MPAN products via reactive uptake to aerosol

water was implemented in CMAQ (Pye et al., 2013). The aqueous chemical mechanism does not necessarily differ between aerosol and cloud water, but certain reactions may be more important in the different regimes (i.e., concentrated versus dilute) (McNeill 2015; Hermann et al., 2015). Here we explore the impacts of including the aqueous IEPOX/MPAN SOA reaction pathway in cloud water and apply the model to an eastern US domain coinciding with SOAS. The platform/period is well-



suited for the task due to the availability of frequent 2-methyltetrol and 2-MG measurements, products of the aforementioned reaction pathways (Table S3), at multiple sites.

Figure 6 shows the modeled June 2013 average SOA increases due to including IEPOX/MPAN chemistry in cloud droplets.
Each panel gives an absolute or percentage concentration difference of impacted SOA species between the extended chemistry (AQCHEM-KMTI) and standard chemistry (AQCHEM-KMT) simulations. In the areas of highest baseline concentrations of IEPOX/MPAN SOA, cloud chemistry contributes an additional ~5-13% to the June average concentration, with a maximum increase in average IEPOX/MPAN SOA of ~20 ng/m$^3$ (Figure 6a). Total SOA from IEPOX/MPAN here is a sum of 2-methyltetrols, 2-MG, organosulfates, organonitrates, and dimers. There is a more significant relative impact (i.e., percentage
increase as opposed to absolute mass change) on 2-MG (Figure 6d) than 2-methyltetrols (Figure 6c). Note, however, that 2-MG concentrations are an order of magnitude lower than those of methyltetrols. The impact on organosulfate (OS), organonitrate (ON), and dimer concentrations with the additional cloud chemistry is negligible, supporting previous work suggesting that OS formation would be minimal at the dilute conditions characteristic of cloud droplets (McNeill et al., 2012). The additional cloud SOA chemistry leads to an average increase of ~10-20% in surface level "cloud SOA" (i.e., SOA formed
within CMAQ cloud water) in the eastern United States (Figure 6b). The largest relative increases in cloud SOA occur, as expected, in areas with periods of persistent cloud cover and high liquid water content during the modeling period (e.g., Lake Michigan).

For the area inside the blue rectangle in Figure 6a, the average vertical concentration differences due to the additional
IEPOX/MPAN cloud chemistry are shown in Figures 7 and 8. There is a 14-16% increase in average cloud SOA concentrations from the surface to 3 km, with a peak percentage increase just above 2 km. This translates to a spatially/temporally averaged IEPOX/MPAN SOA concentration increase of 8 ng/m$^3$ in the layers closest to the surface (Figure 8). Larger impacts can be seen during shorter timescales at locations that are characterized by the availability of both adequate precursor levels and cloud liquid water content (Figure 11).

PM filters were analyzed for the tracer compounds, 2-methyltetrols and 2-MG, at select sites during the SOAS field campaign (Budisulistiorini et al., 2015). A comparison of modeled and measured concentrations of these species at the RTP site in NC (where some of the largest modeled differences occur of the available measurement sites) are given in Figure 9. While the impacts of the additional cloud chemistry at RTP during this period are not very large, there are small increases in 2-
methyltetrol and 2-MG predictions. This leads to slightly better error statistics due to the fact that the base model tends to underpredict the concentrations of these compounds at the RTP site (Table 3). While the additional cloud chemistry does not drastically increase concentrations at RTP, there are areas where modeled concentration impacts can be significant (Figure 10). To investigate under what conditions the largest contributions from cloud-water production of IEPOX/MPAN SOA might be observed, we extracted the time-series of products and precursors at the cell with the largest hourly impact during the June





2013 simulation in Figure 11. Figure 11 shows the hourly $\Delta SOA_{IEPOX/MPAN}$, $\Delta SOA_{2-MG}$, liquid water content, and precursor species IEPOX and MAE for the max difference cell. The additional chemistry included here does not lead to a temporally-uniform change in concentrations, with impacts from additional cloud SOA only discernible for sporadic spikes during the latter part of the month. The large SOA impacts coincide with periods of simultaneously high liquid water content and high

precursor concentrations. A large fraction of the total increase in $SOA_{IEPOX/MPAN}$ (red line) is due to increases in 2-MG production (grey line) even though 2-methyltetrols dominate the concentration of total $SOA_{IEPOX/MPAN}$ in most of the domain. If the Henry's law coefficient for IEPOX is much higher than used here (for example, $1.7 \times 10^8$, Gaston et al. (2014)) then aqueous processing of IEPOX would lead to even higher methyltetrol concentrations than predicted here.

One of the more uncertain aspects of this chemistry may be the production of SOA from the MPAN products. In the current model formulation, 2-MG production is mainly based on the physical constants/chemical reaction rates for MAE. However, other studies have indicated that HMML may be a more dominant precursor to 2-MG (Nguyen et al., 2015). Additionally, we assume that the products of these aerosol and cloud aqueous pathways are non-volatile, while monomeric species like 2-methyltetrols and 2-MG may be better represented as semi-volatile (Isaacman-VanWertz et al., 2016). These are areas that

may be worthwhile to further refine in future development efforts, as more laboratory/field research becomes available, in an effort to better quantify the relative impacts of cloud versus aerosol water production pathways on SOA mass.

## 4 Summary and future directions

We have developed a framework for extending CMAQ cloud chemistry and have implemented two additional cloud chemistry options, AQCHEM-KMT and AQCHEM-KMTI, in CMAQv5.1. While CMAQ's standard cloud chemistry module

(AQCHEM) is structurally limited to a simple chemical mechanism, this work advances our ability to implement and investigate additional aqueous chemical pathways and the impacts of microphysical parameters on cloud chemistry in CMAQ.

KPP was used to generate a Rosenbrock solver to integrate the stiff system of ODEs that describe the mass transfer, chemical kinetics, and scavenging processes of CMAQ clouds. Box model tests were performed to choose efficient solver and tolerance

settings, validate the code generated with KPP, and examine the potential impacts of the new solver and mass transfer limitations on model concentrations of $SO_4^{2-}$ and SOA. Month-long winter and summer CMAQ simulations over CONUS reveal that while the new solver and mass transfer considerations do not cause large changes in regional or long-term average concentrations for the standard aqueous mechanism, more significant impacts are observed on shorter timescales. The addition of in-cloud SOA production from IEPOX/MPAN species led to an average increase in cloud SOA concentrations of around

15% for June 2013 and slightly improved error and bias statistics for IEPOX/MPAN SOA at the RTP measurement site during the SOAS field campaign period.



While the new solver and kinetic mass transfer treatment produced sporadic and small differences on average for the standard cloud chemistry mechanism, a significant value in AQCHEM-KMT is that with the automatic code generation of KPP, the modeling framework provides a straightforward way to implement and test new chemical pathways and determine the sensitivity of model concentrations to uncertain parameters and representations. Including kinetic mass transfer treatment

allows cloud chemistry, wet deposition, and the distribution of species between phases to be more directly affected by microphysical parameters like droplet size. Adding linkages between cloud chemistry and microphysical parameters (e.g., cloud droplet radius or activated aerosol fraction) allows for a better representation of feedback between clouds, aerosols, and radiation, increasing our ability to determine what linkages are most influential on the model species or processes of greatest interest.

While incremental improvements in the computational efficiency occurred throughout the development process of AQCHEM-KMT, it can contribute to longer CMAQ run times compared to AQCHEM (6-35% for the scenarios investigated here). Chemical transport models like CMAQ require a balance between accuracy and efficiency, and that balance often depends on the goal of a particular model application. For applications geared towards estimating $PM_{2.5}$ over monthly or seasonal

averaging times, for example, equilibrium assumptions likely will not lead to significant errors *for the standard AQCHEM mechanism*. If the focus is on individual aerosol species over shorter time scales or currently unrepresented or parameterized chemistry, however, those assumptions in AQCHEM might lead to an undesired loss in accuracy, and AQCHEM-KMT might be the preferred choice. Future implementations of extended cloud chemistry in CMAQ should continue to strive towards greater computational efficiency. Alongside efforts to expand the cloud chemical mechanism to include the most

relevant/impactful chemical pathways, special efforts should also be made to improve the module efficiency, including the application of simplifying equilibrium assumptions for individual species or conditions as appropriate.

**Code availability**

CMAQ model documentation and CMAQ version 5.1 source code, including AQCHEM, AQCHEM-KMT, and AQCHEM-

KMTI cloud chemistry options, are available at www.cmaq-model.org.

**Disclaimer**

Although this paper has been subjected to US EPA review and approved for publication, the views and interpretations in this publication are those of the authors and not necessarily reflective of their organizations' policies or views.

**Acknowledgments**

We would like to thank Jon Pleim, Deborah Luecken, and Shawn Roselle for constructive comments on our manuscript, as well as Rohit Mathur, Neha Sareen, Jason Surratt, and Barbara Turpin for helpful discussion during the course of this work.



The contributions of C.O.S. and J.B. were supported, in part, by grant RD-83386501-0 from the Environmental Protection Agency.

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



**Table 1: Comparison of cloud chemistry models included in CMAQv5.1.**

| Process | AQCHEM standard | AQCHEM-KMT | AQCHEM-KMTI |
|---|---|---|---|
| Solver | Forward Euler | Rodas3 (Rosenbrock) | Rodas3 (Rosenbrock) |
| Gas-aqueous mass transfer | Henry's Law Equilibrium | Kinetic mass transfer with gas diffusion/interfacial limitation | Kinetic mass transfer with gas diffusion/interfacial limitation |
| Ionic dissociation | Equilibrium | Forward/reverse reactions | Forward/reverse reactions |
| Chemistry | 5 S(IV) to S(VI) oxidation reactions + 2 SOA forming reactions from glyoxal and methylglyoxal | Same mechanism as AQCHEM. Includes aqueous diffusion correction for $O_3$ | AQCHEM mechanism + IEPOX/MPAN chemistry. Includes aqueous diffusion correction for $O_3$, and IEPOX/MPAN |
| pH | [H$^+$] estimated at each time step using a bisection method while maintaining electroneutrality. | Dynamic [H$^+$]. Initial value is based on activated aerosol ions at t=0 assuming electroneutrality | Dynamic [H$^+$]. Initial value is based on activated aerosol ions at t=0 assuming electroneutrality |
| Other | Instantaneous activation of all accumulation and coarse mode species. | Instantaneous activation of all accumulation and coarse mode species. | Instantaneous activation of all accumulation and coarse mode species. |

**Table 2: Modeled processes and associated rate coefficients and equations.**

| Process | Equations | Rate coefficients | Other information* |
|---|---|---|---|
| Gas-Liquid phase transfer | $C_{g,i} \xrightarrow{k_f} C_{aq,i}$ | $k_f = k_{mt,i} w_L$ | $k_{mt,i}\left(s^{-1}\dfrac{vol_{air}}{vol_{H2O}}\right) = \left(\dfrac{r^2}{3D_{g,i}} + \dfrac{4r}{3\bar{v}_i \alpha_i}\right)^{-1}$ |
| Liquid-Gas phase transfer | $C_{aq,i} \xrightarrow{k_b} C_{g,i}$ | $k_b = \dfrac{k_{mt,i}}{H_{T,i}RT}$ | $\bar{v}_i = \sqrt{\dfrac{8RT}{MW_i\pi}}$ |
| X$_{ion,f}$: Dissociation | $C_{aq,i} \xrightarrow{k_f} C_{aq,i}^{-1} + H^+$ | $k_b$ = literature value, T independent | $Keq_{i,T} = Keq_{i,Tref}\left[\dfrac{-\Delta H_a}{R}\left(\left(\dfrac{1}{T}\right) - \left(\dfrac{1}{T_{ref}}\right)\right)\right]$ |
| X$_{ion,b}$: Association | $C_{aq,i}^{-1} + H^+ \xrightarrow{k_b} C_{aq,i}$ | $k_f = Keq_{i,T}k_b$ | Activity coefficients are rolled into the forward and backward rates as appropriate |
| A$_{scav}$: Droplet scavenging of interstitial aerosol | $C_{aer,i,akn} \xrightarrow{\alpha} C_{aq,i}$ | α | α is the attachment rate for interstitial aerosols, calculated external to the aqueous chemistry module according to Binkowski and Roselle (2003) |
| W$_{dep}$: Wet deposition | $C_{aq,i} \xrightarrow{W_{dep}} C_{WD,i}$ | $W_{dep} = \dfrac{1}{\tau_{wash}}$ | $\tau_{wash(s)} = \dfrac{WT_{AVG} \times CTHK \times 3600}{PRATE}, 0$ where WT$_{AVG}$ = total liquid water content, CTHK = cloud thickness, PRATE = precipitation rate |
| R$_{aq}$: Chemical kinetics | $C_{aq,1} + C_{aq,2} \xrightarrow{k_{rxn}} C_{aq,3}$ | $k_{rxn}$ | Complex rate coefficients are set according to the CMAQ base mechanism (Sarwar et al., |



| | | | | | | |
|---|---|---|---|---|---|---|
2013; Carlton et al., 2010). While most droplet species are assumed to be well-mixed, a correction factor may be applied to account for aqueous diffusion limitations (Table S3). This correction factor, $Q_i$, relates surface and bulk droplet concentrations.

$$k_{rxn,effective} = k_{rxn} Q_i$$

$$Q_i = 3\left(\frac{coth(q_i)}{q_i} - \frac{1}{q_i^2}\right), Q_i \leq 1$$

$$q_i = r\sqrt{\frac{k_i}{D_{aq,i}}}$$

*MW is the molecular weight, $D_{g,i}$ is the gas phase diffusion coefficient (m²/s), $\alpha_i$ is the accommodation coefficient, $q_i$ is the diffuso-reactive parameter, $D_{aq,i}$ is the aqueous phase diffusion coefficient (m²/s), and $k_i$ is the effective first order reaction rate of species i. r is the droplet radius (m).

5  **Table 3: Observation-prediction statistics for 2-methyltetrols and 2-MG at the RTP measurement site during SOAS.**

| | | Base | New | | Base | New |
|---|---|---|---|---|---|---|
| Normalized Mean Bias | 2-MG | -88.2 | -78.6 | 2-methyltetrols | -58.6 | -54.8 |
| Normalized Mean Error | | 91.1 | 83.0 | | 79.2 | 77.5 |
| Correlation | | 0.21 | 0.29 | | 0.16 | 0.17 |
| Model avg (ng/m³) | | 1.03 | 1.86 | | 56.8 | 61.9 |




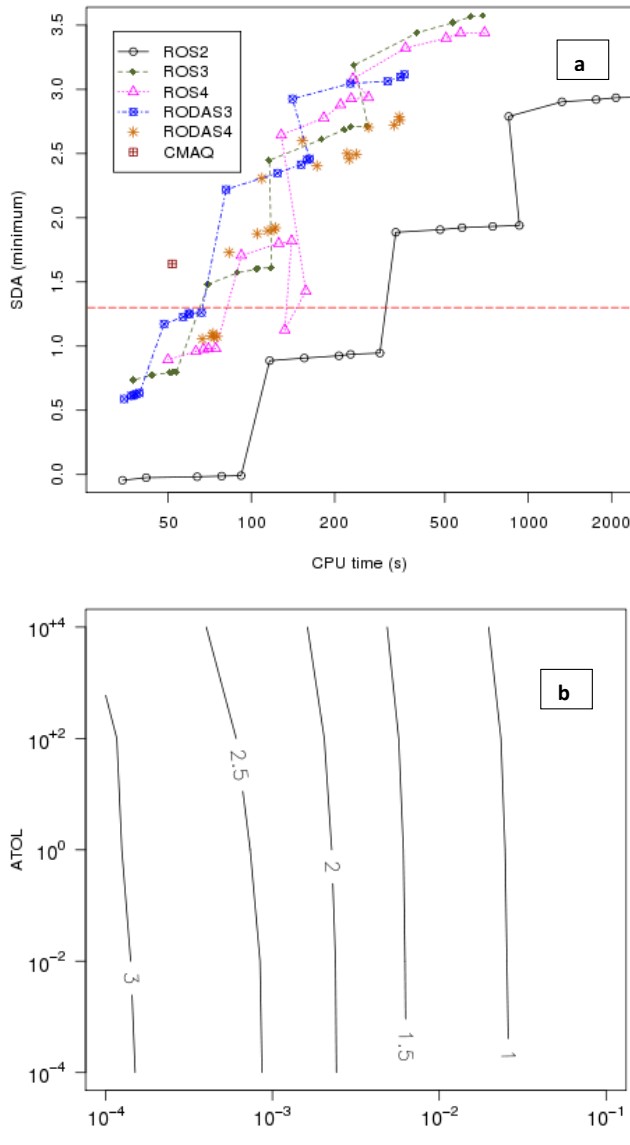

5    **Figure 1: Significant digits of accuracy (SDA) for the CMAQ species with the maximum error for (a) different variants of Rosenbrock solvers and (b) the Rodas3 solver at different combinations of relative and absolute tolerance.**





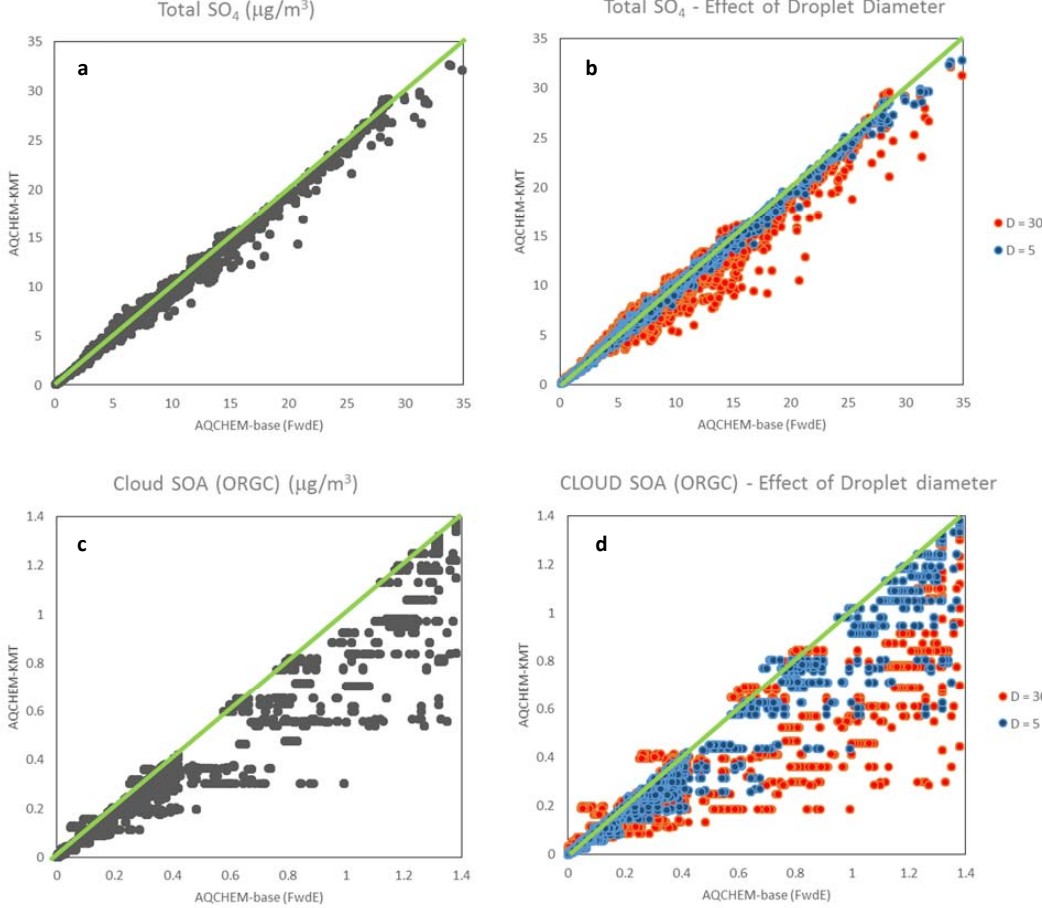

**Figure 2: AQCHEM-KMT vs. standard AQCHEM predictions for (a) total SO$_4^{2-}$ (sum over all modes) and (c) SOA from cloud processing of carbonyls (ORGC) at default droplet diameter of 16 μm as well as at 5 and 30 μm for (b) total SO$_4^{2-}$ and (d) ORGC.**




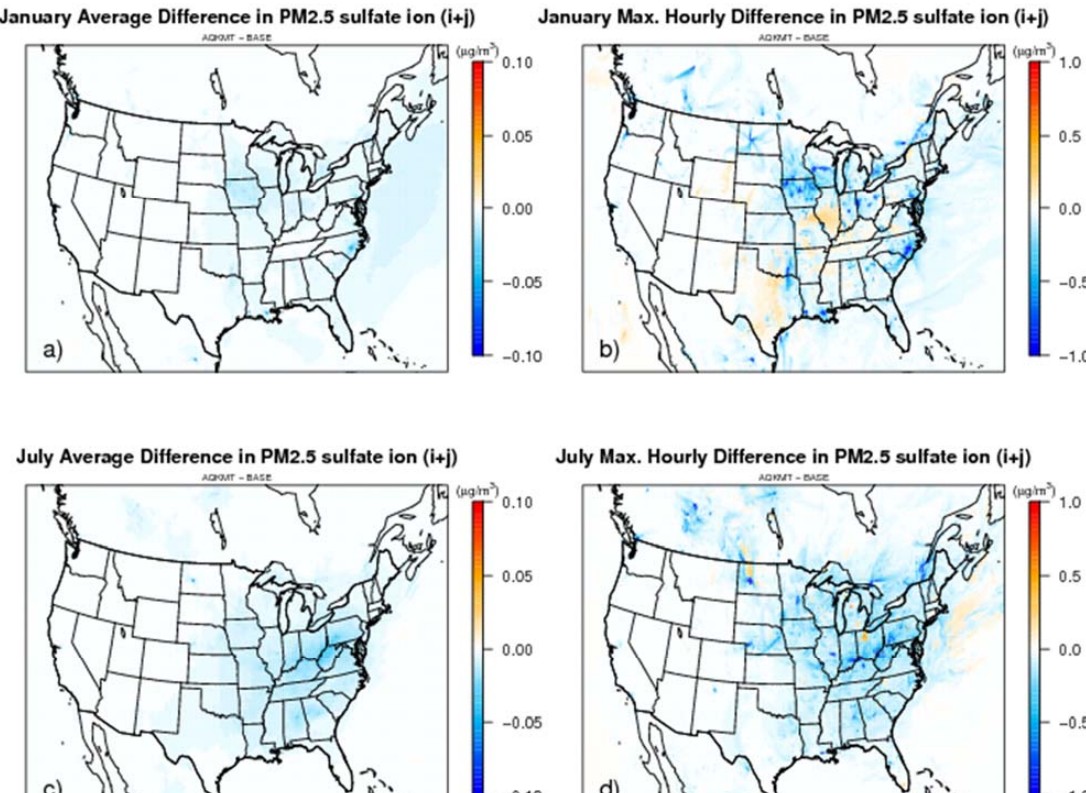

**Figure 3: Average and maximum hourly difference (AQKMT - Base) in fine SO$_4^{2-}$ (µg/m$^3$) for (a) January 2011 (average), (b) January 2011 (maximum), (c) July 2011 (average) and (d) July 2011 (maximum) using CMAQv5.1. Note the different scales for average and maximum plots.**



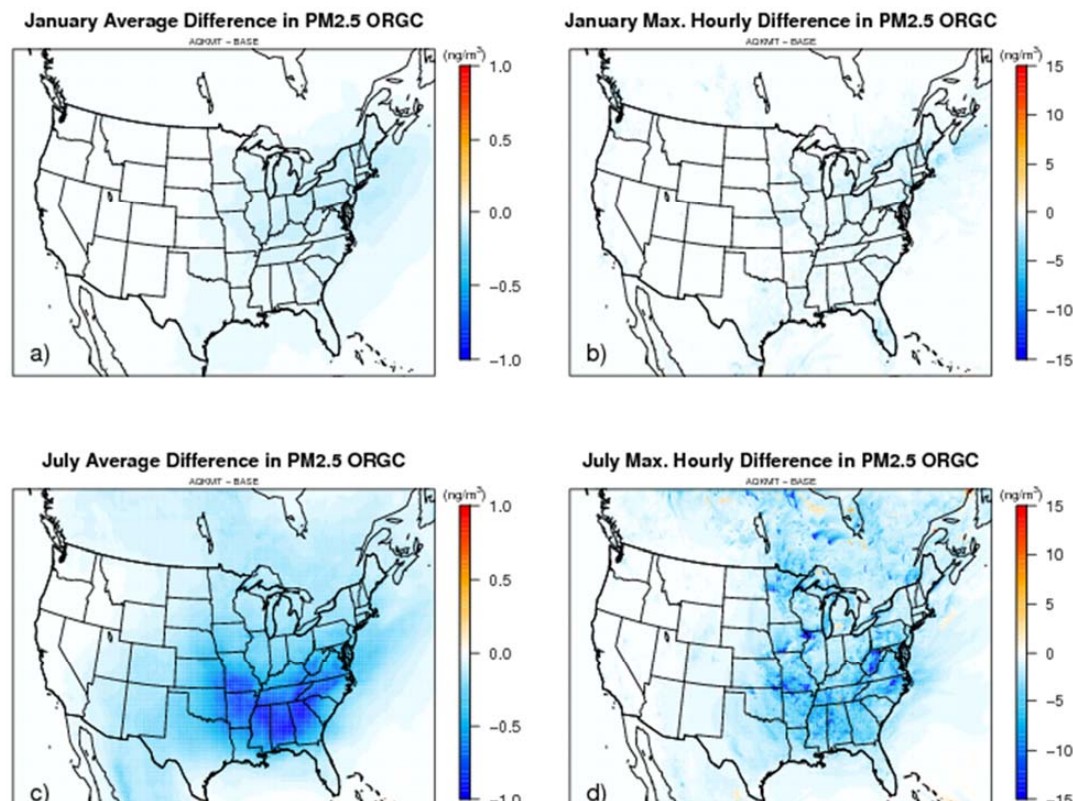

**Figure 4: Average and maximum hourly difference (AQKMT - Base) in SOA from cloud processing of carbonyls (ng/m³) for (a) January 2011 (average), (b) January 2011 (maximum), (c) July 2011 (average) and (d) July 2011 (maximum) using CMAQv5.1**





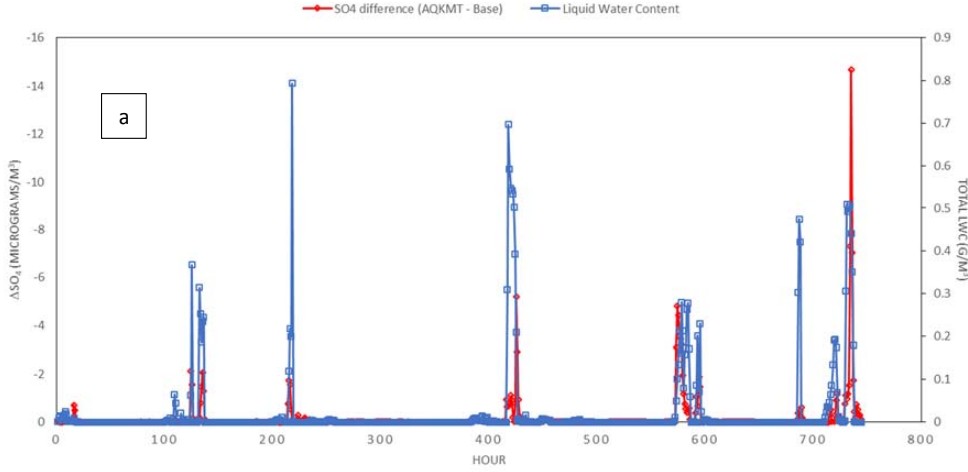

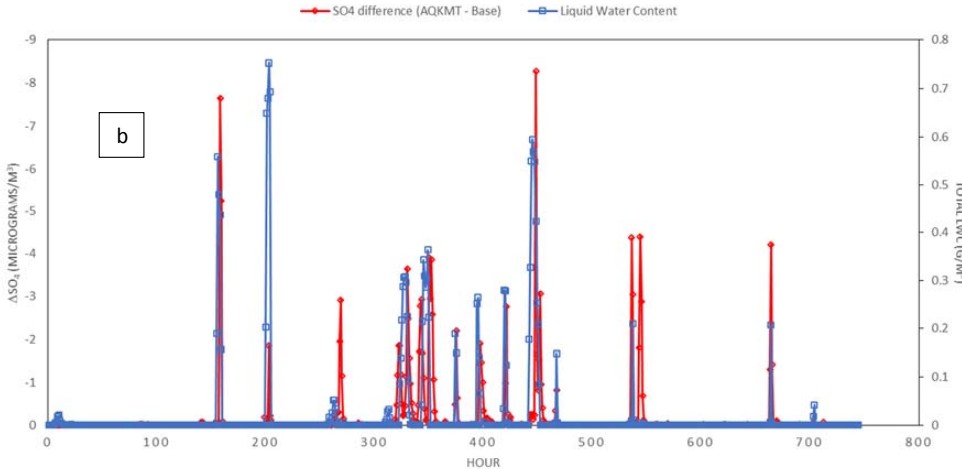

**Figure 5: Modeled hourly liquid water content (blue, g/m³) and change in fine SO₄²⁻ (red, µg/m³) (KMT-Base) in the cell containing the maximum (absolute) hourly difference for (a) January and (b) July 2011.**





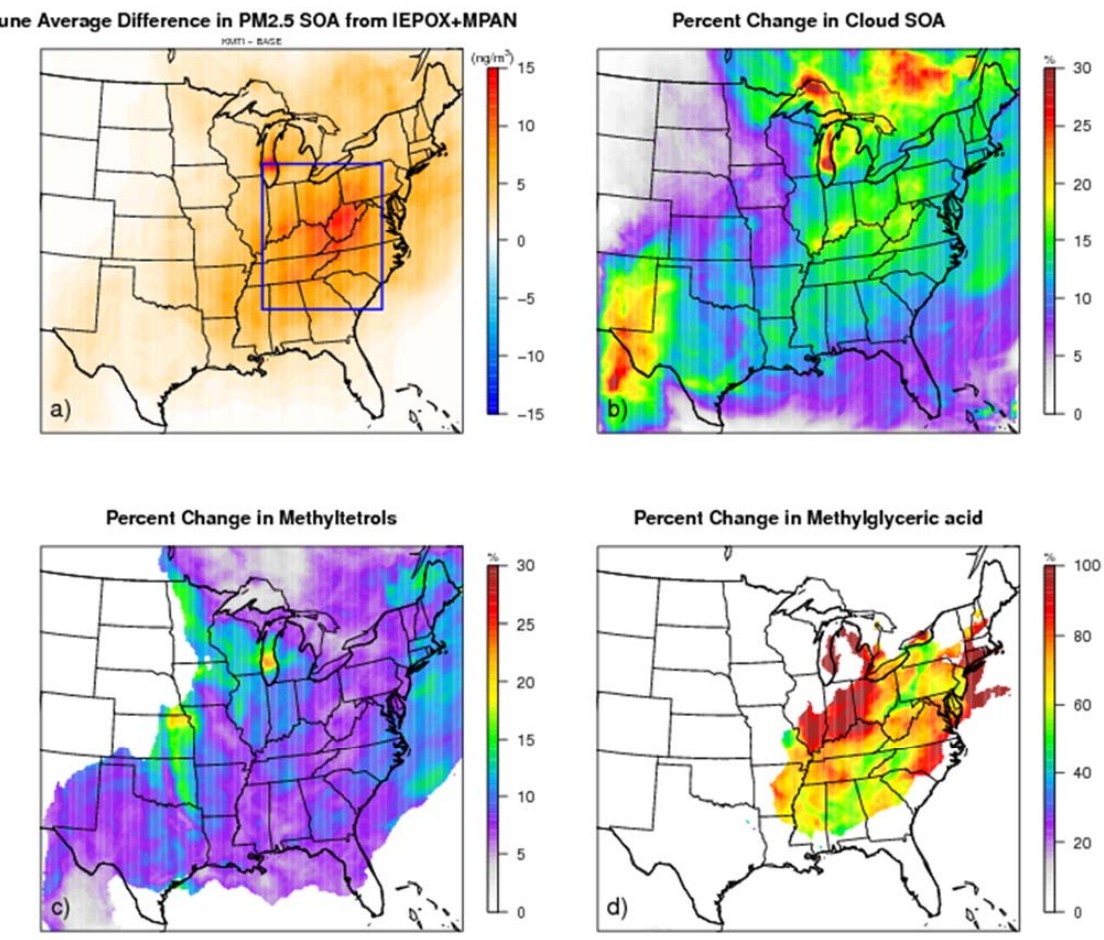

**Figure 6: (a) June 2013 average increase in SOA from IEPOX/MPAN and percentage increase in surface level (b) "cloud SOA" (> 1 ng/m³), (c) 2-methyltetrols (> 10 ng/m³), and (d) 2-MG (> 1 ng/m³). "SOA from IEPOX+MPAN" is the sum of 2-methyltetrols, 2-MG, and related organosulfates, organonitrates, and oligomers.**




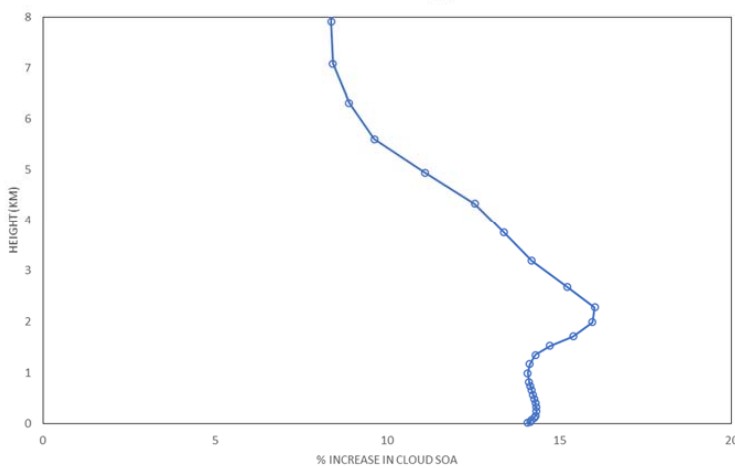

**Figure 7: June 2013 average vertical distribution of the estimated percentage increase in "cloud SOA" with the addition of SOA from IEPOX/MPAN in cloud water. These values are averaged spatially over the area indicated by the blue box in Figure 6a.**

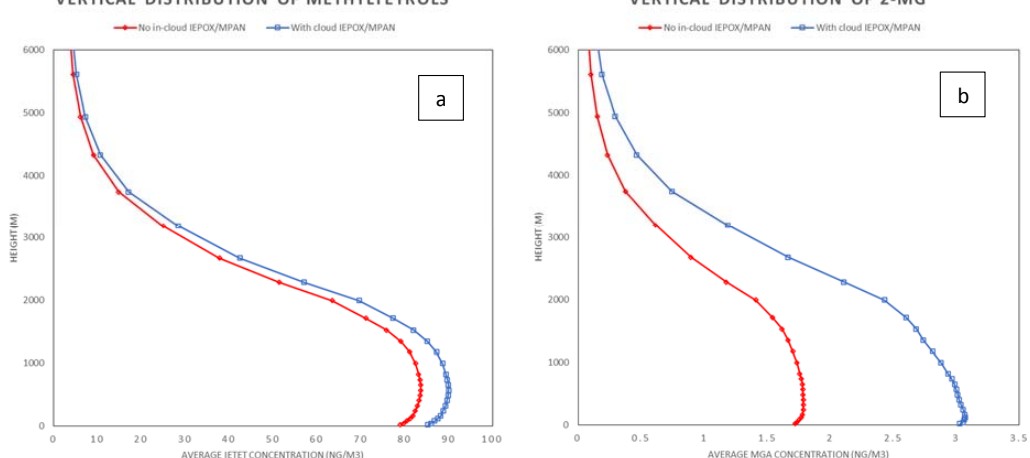

**Figure 8: June 2013 vertical distribution of (a) 2-methyltetrols (ng/m$^3$) and (b) 2-MG (ng/m$^3$), averaged spatially over the area indicated by the blue box in Figure 6a.**



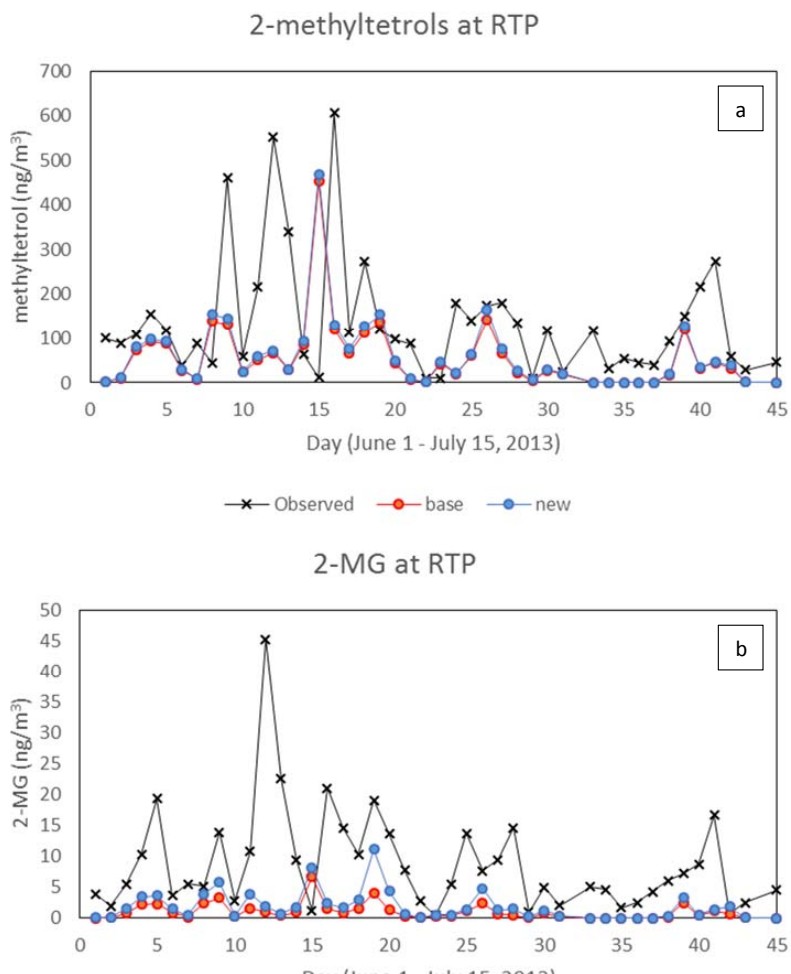

**Figure 9: Observed and modeled (a) 2-methyltetrols (ng/m³) and (b) 2-MG (ng/m³) at Research Triangle Park, NC, during June 1-July 15, 2013, for a base (red) simulation and a simulation with additional in-cloud formation of IEPOX/MPAN SOA (blue).**





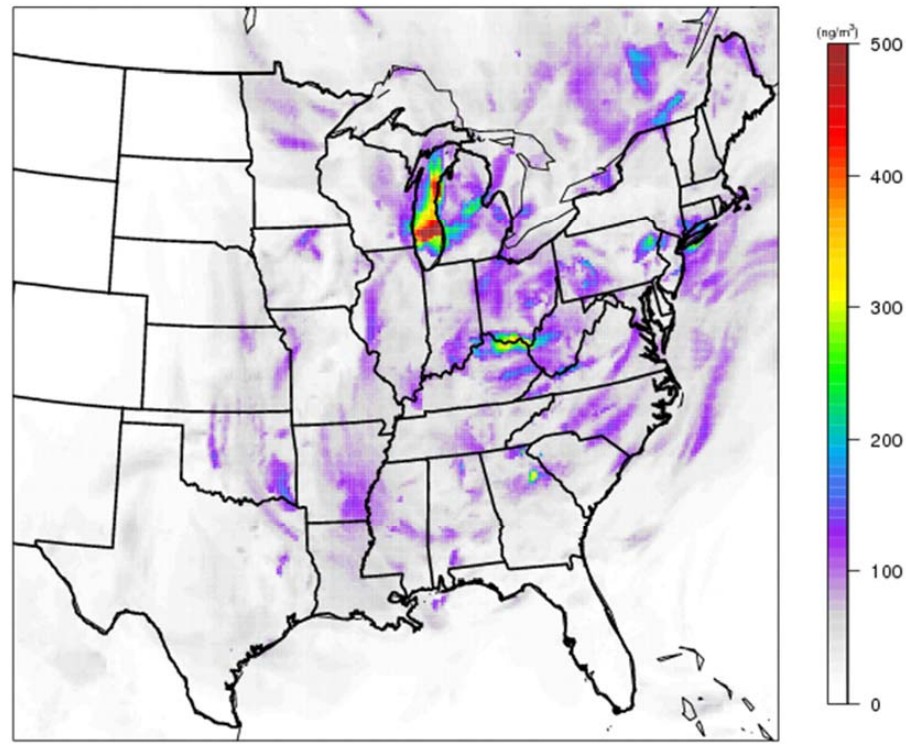

**Figure 10: Maximum increase in hourly IEPOX/MPAN SOA ([AQCHEM-KMTI] – [AQCHEM-KMT]) (ng/m³) for June 2013.**





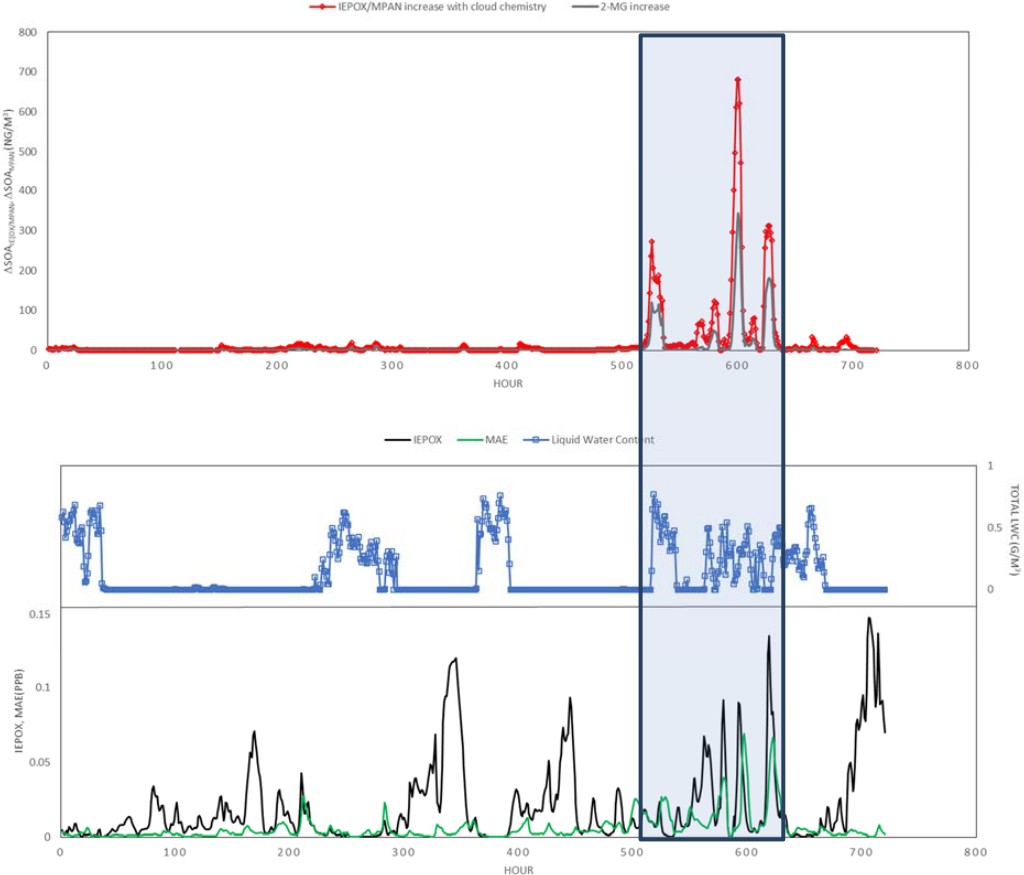

**Figure 11: Change in SOA$_{IEPOX/MPAN}$ due to in-cloud production (alongside precursor and liquid water content levels) for the cell containing the highest hourly difference during June 2013. Shown from top to bottom are (top) the change in predicted hourly concentrations of total SOA$_{IEPOX/MPAN}$ (ng/m$^3$) (red) and 2-MG (ng/m$^3$) (grey) with the additional in-cloud SOA production, (middle) cell liquid water content (g/m$^3$) (blue), and (bottom) IEPOX (ppb) (black) and MAE (ppb) (green) concentrations. Between 500-650 hours, on average more than 40% of the additional SOA was 2-MG.**