# Peer review of "A framework for expanding aqueous chemistry in the Community Multiscale Air Quality (CMAQ) model version 5.1"

_Geoscientific Model Development, 2016_

## Referee Comment (RC1) · Anonymous Referee #1 · 2 Jan 2017

Kahey et al. replace the hard-coded aqueous chemistry mechanism in CMAQ with a flexible chemical mechanism using the KPP solver. This allows a more general approach, with aspects such as mass transfer and H+ concentration now allowed to evolve dynamically. The authors demonstrate this method by first replicating the existing AQCHEM mechanism (AQCHEM-KMT), and then extending it with a more complete representation of organic chemistry (AQCHEM-KMTI). This demonstration shows little impact on the model representation of sulfur chemistry, but secondary organic aerosol is significantly affected. Their approach sidesteps some issues of the previous sulfur-focused implementation, such as the application of inappropriately long timesteps to the calculation of rapidly-evolving SOA concentrations, and delivers a useful tool for other groups to explore other aspects of aqueous phase chemistry.

This paper represents an incremental development in the representation of aqueous phase chemistry in a regional-scale model. However, such a development, especially in a model as widely used as CMAQ, is both overdue and welcome. The reapplication of KPP to solve aqueous phase chemistry seems natural and well implemented, and I am pleased to recommend this paper for publication in GMD, pending some minor revisions. I have outlined two general criticisms below, followed by a list of minor recommendations.

My first and most significant issue relates to the question of computational efficiency. The final paragraph of the conclusions is informative, evenly discussing the computational trade-offs associated with the new, more complete aqueous chemistry mechanism. However, it references timing data which has not been given previously. Section 2.1 or 2.2 would benefit significantly from a dedicated discussion of the execution time of AQCHEM in each of the scenarios in comparison to AQCHEM-KMT. It would also be useful to know the degree to which some of the assumptions previously used by AQCHEM were found to be inaccurate. For example, with AQCHEM-KMT, the authors can diagnose the (in)accuracy of the electroneutrality assumption previously forced on the aerosol.

My second issue regards operator splitting. Previous work has shown that splitting gas- and aerosol-phase chemistry into two separate operations inevitably introduces errors, and that these errors can be quite large (Djouad and Michelangeli, 2004). This is especially pertinent now that mass transfer limitations are being explicitly considered. Unfortunately Kahey et al. do not state how their aqueous chemistry operator fits into CMAQ as a whole. My assumption is that it remains a separate operation from gas-phase chemistry, but without a clear explanation the reproducibility of the work is compromised. The paper would benefit from a description of the order of operation of AQCHEM-KMT with respect to the rest of the model operators.

**Specific comments and technical corrections**

The following minor issues should be addressed:

- Page 5, line 18: The sentence beginning "While in AQCHEM..." is confusing and would benefit from a rewrite.

- Page 8, line 13: Suggest that "Figures 2 b and d" is changed to "Figures 2b and 2d"

- Page 11, line 7: The phrasing "At least a couple micrograms per meter" is rather vague.

- Page 11, first paragraph: Some sense should be given of the relative impact of implementing AQCHEM-KMT. With only absolute differences given, it is difficult to tell if the changes are ever significant. Suggest the authors state what the maximum and average percentage change in the given grid cell is. Similarly, Figures 3 and 4 would benefit from an additional panel showing the "baseline" (AQCHEM) concentrations in January and July. Without this, it is difficult to tell the significance of AQCHEM-KMT's changes.

- Figure 1a: The caption is unclear. Specifically, it should be stated exactly what is varying between different points for the same solver (ie that different tolerances are being tested, and over what range). It would also be helpful to point out explicitly what is changing between the different plateaus - it appears that SDA is more sensitive to relative tolerance than to absolute tolerance, based on the clustering behavior.

- All of the figures showing land area should have axis markers (latitutde and longitude, if possible, or at least the X and Y dimension indices). Once these are in place, the grid index given on page 11, line 5 should be changed to match the

dimensions used for the figures, so that the reader can identify which point on the grid is being discussed.

- Table 2: There are some typesetting issues here, particularly where a subscript should or should not have been used. Note, for example, the "Other information" entry for wet deposition.

**References**

Djouad, R. and Michelangeli, D. V.: Investigation of splitting gas and aqueous operators in atmospheric multiphase box models, Atmos. Res., 71(4), 253–263, 2004.

---

## Referee Comment (RC2) · Anonymous Referee #2 · 6 Jan 2017

Fahey et al. developed a new cloud chemistry mechanism (AQCHEM-KMT) for use in large-scale models and implemented it into CMAQ. AQCHEM-KMT allows for the investigation of aqueous-phase chemical reactions that could not be implemented into the aqueous-phase chemistry mechanism that is currently implemented in CMAQ (AQCHEM) because of its explicit consideration of processes such as mass transfer limitations. As most large-scale models use an aqueous-phase chemical mechanism similar to AQCHEM, this development represents a significant step forward in examining the influence of aqueous-phase chemical reactions on the chemical composition of the atmosphere in large-scale models. The paper is well written, and I recommend publication in GMD after some very minor issues are addressed.

The authors find much larger differences between AQCHEM-KMT and AQCHEM for sulfate than for SOA, but the reasons for this difference could be more explicitly discussed than they currently are. What other aqueous-phase reactions or types of reactions can now be implemented in AQCHEM-KMT that was not possible with AQCHEM? Are there other sulfate production mechanisms that are better suited for AQCHEM-KMT than AQCHEM that are not currently included in the model, or do the authors expect sulfate to almost always be similar for AQCHEM-KMT and AQCHEM on monthly time scales?

In the introduction, it would be good to specify that when you refer to the aqueous phase you are referring specifically to cloud droplets and not liquid water associated with aerosols.

Page 3 line 13: Refer to Table S3 so one can readily find the seven oxidation reactions, or list them here.

Many abbreviations/acronyms in the text and tables are not defined (e.g., MPAN). Perhaps add a table of abbreviations/acronyms in the SI.
* * *

---

## Referee Comment (RC3) · Anonymous Referee #3 · 19 Jan 2017

Fahey et al. presents an implementation of two new cloud aqueous phase chemistry options (AQCHEM-KMT and AQCHEM-KMTI) for the CMAQ model. The paper is well written and will be of interest to the readers of GMD, I therefore support its publication in GMD after issues raised Referees #1 and #2 have been addressed.

Minor issues: - Page 9, Line 29: Should "in-line calculation of photolysis rates" be "online calculations of photolysis rates"?

---

## Author Comment (AC1) · 2 Mar 2017

Response to reviewer #1

**The authors thank reviewer #1 for their comments and suggested revisions. In the following, we restate the reviewer comments and follow with our response in "bold" font. Note that any references to page/line number are made for both the "updated" manuscript (with markup) and the original discussion paper.**

Kahey et al. replace the hard-coded aqueous chemistry mechanism in CMAQ with a flexible chemical mechanism using the KPP solver. This allows a more general approach, with aspects such as mass transfer and H+ concentration now allowed to evolve dynamically. The authors demonstrate this method by first replicating the existing AQCHEM mechanism (AQCHEM-KMT), and then extending it with a more complete representation of organic chemistry (AQCHEM-KMTI). This demonstration shows little impact on the model representation of sulfur chemistry, but secondary organic aerosol is significantly affected. Their approach sidesteps some issues of the previous sulfur-focused implementation, such as the application of inappropriately long timesteps to the calculation of rapidly-evolving SOA concentrations, and delivers a useful tool for other groups to explore other aspects of aqueous phase chemistry.

This paper represents an incremental development in the representation of aqueous phase chemistry in a regional-scale model. However, such a development, especially in a model as widely used as CMAQ, is both overdue and welcome. The reapplication of KPP to solve aqueous phase chemistry seems natural and well implemented, and I am pleased to recommend this paper for publication in GMD, pending some minor revisions. I have outlined two general criticisms below, followed by a list of minor recommendations.

My first and most significant issue relates to the question of computational efficiency. The final paragraph of the conclusions is informative, evenly discussing the computational trade-offs associated with the new, more complete aqueous chemistry mechanism. However, it references timing data which has not been given previously. Section 2.1 or 2.2 would benefit significantly from a dedicated discussion of the execution time of AQCHEM in each of the scenarios in comparison to AQCHEM-KMT.

**We have added the following text to section 2.2, p 9, line 13 (p 8, line 28 in original document):**

**It should be noted that by introducing the new solver and relaxing equilibrium assumptions that the computational requirements of AQCHEM-KMT significantly exceed those of AQCHEM, even before adding new chemical species or reactions. On average AQCHEM can simulate the scenarios of Table S5 with a runtime on the order of ~1 second, while AQCHEM-KMT requires ~65 seconds to model the scenario set. While cloud chemistry only accounts for a fraction of the computational time required by a three-dimensional chemical transport model like CMAQ, implementation of AQCHEM-KMT in a chemical transport model (CTM) will lead to an overall increase in CTM run time that will vary depending, in part, on the cloudiness of a modeled period. These requirements will likely increase as the chemical mechanism expands, and future efforts should be dedicated to investigating how to make the model more efficient, including revisiting equilibrium assumptions for certain processes or species.**

It would also be useful to know the degree to which some of the assumptions previously used by AQCHEM were found to be inaccurate. For example, with AQCHEM-KMT, the authors can diagnose the (in)accuracy of the electroneutrality assumption previously forced on the aerosol.

**We have added the following text to section 2.2, p 8, line 29 (p 8, line 23 in original document):**

**Additional box modeling investigations with a slightly expanded mechanism indicate that when one ignores aqueous diffusion limitations (for the default droplet diameter of 16 μm), assumes instantaneous equilibrium for ionic dissociation, and calculates pH assuming electroneutrality (but maintains the kinetic mass transfer treatment for transfer between the phases) that the predicted concentrations of $SO_4^{2-}$, ORGC, and other major species are comparable to those predicted with the fully dynamic approach of AQCHEM-KMT. This indicates that the largest differences between AQCHEM-KMT and AQCHEM for the test scenarios are a result of using kinetic mass transfer coefficients to describe the transfer of species between the phases (i.e., not assuming instantaneous Henry's law equilibrium) and to a lesser extent the change in solvers.**

My second issue regards operator splitting. Previous work has shown that splitting gas and aerosol-phase chemistry into two separate operations inevitably introduces errors, and that these errors can be quite large (Djouad and Michelangeli, 2004). This is especially pertinent now that mass transfer limitations are being explicitly considered. Unfortunately Kahey et al. do not state how their aqueous chemistry operator fits into CMAQ as a whole. My assumption is that it remains a separate operation from gas-phase chemistry, but without a clear explanation the reproducibility of the work is compromised. The paper would benefit from a description of the order of operation of AQCHEM-KMT with respect to the rest of the model operators.

**The following text has been added to page 10, line 13 (p 9, line 19 in original document): "Note that in all CMAQ cases, cloud chemistry and gas phase chemistry are not solved simultaneously but are instead solved in separate operators. Following advection and diffusion, cloud processes (including cloud chemistry) are treated for resolved and sub-grid clouds. This is followed by gas phase chemistry (including heterogeneous chemistry on aerosols) and aerosol dynamics. Inevitably there are errors that can result from estimating the impacts of chemistry of different phases separately, and in the future, the feasibility of simultaneously solving chemistry across all phases will be investigated."**

Specific comments and technical corrections
The following minor issues should be addressed:
• Page 5, line 18: The sentence beginning "While in AQCHEM..." is confusing and would benefit from a rewrite.

**The sentence now reads as follows: "While instantaneous Henry's Law equilibrium is assumed for all species in AQCHEM, in actuality, the distribution of a species between the gas and aqueous phases may deviate significantly from equilibrium"**

• Page 8, line 13: Suggest that "Figures 2 b and d" is changed to "Figures 2b and 2d"

 **"Figures 2 b and d" has been changed to "Figures 2b and 2d"**

• Page 11, line 7: The phrasing "At least a couple micrograms per meter" is rather vague.

**Changed "concentrations can be at least a couple micrograms per cubic meter in magnitude" to "concentrations often exceed 2 micrograms per cubic meter in magnitude"**

• Page 11, first paragraph: Some sense should be given of the relative impact of implementing AQCHEM-KMT. With only absolute differences given, it is difficult to tell if the changes are ever significant. Suggest the authors state what the maximum and average percentage change in the given grid cell is. Similarly, Figures 3 and 4 would benefit from an additional panel showing the "baseline" (AQCHEM) concentrations in January and July. Without this, it is difficult to tell the significance of AQCHEM-KMT's changes.

**Figures 3 and 4 have been updated to include the average "baseline" AQCHEM concentrations of $SO_4^{2-}$ and ORGC for January and July. Figure captions have been updated to reflect the additional AQCHEM concentration plots. Section 3.1 has been updated due to the change to the figures as follows (additions/updates are underlined):**

**Page 11, line 26 (p 10, line 26 in original document) is now "Figures 3 and 4 show the average baseline (AQCHEM) concentrations and difference in predictions between AQCHEM (base)…"**

**Page 11, lines 27-28 (p 10, line 27 in original document): "In addition to a map of the average baseline concentrations (a,d), the figures include a map of monthly average (b,e) and maximum hourly (c,f) differences for January (top) and July (bottom) 2011"**

**The following text has also been added on page 12 lines 6-9 (p 11, line 5 in original document) (new text underlined) "The figures also include modeled total liquid water content values. While monthly average $SO_4^{2-}$ predicted with AQCHEM-KMT is only 5.2% and 6.5% lower than the base in the cell selected for January and July respectively, when the maximum hourly difference is observed, AQCHEM-KMT predicts**

**35% less SO$_4^{2-}$ at cell (264,54) and 15% less SO$_4^{2-}$ at cell (183,213) compared to AQCHEM. For most hours,…"**

• Figure 1a: The caption is unclear. Specifically, it should be stated exactly what is varying between different points for the same solver (ie that different tolerances are being tested, and over what range). It would also be helpful to point out explicitly what is changing between the different plateaus - it appears that SDA is more sensitive to relative tolerance than to absolute tolerance, based on the clustering behavior.

**The Figure 1 caption has been changed to read as follows: "Figure 1: Significant digits of accuracy (SDA) for the CMAQ species with the maximum error for (a) different variants of Rosenbrock solvers and (b) the Rodas3 solver at different combinations of relative and absolute tolerance. Each point in figure 1a represents a different relative and absolute tolerance combination. The absolute tolerances tested were 10$^{-4}$, 10$^{-2}$, 10$^{0}$, 10$^{2}$, and 10$^{4}$ molecules/cm$^{3}$ air for relative tolerances = 10$^{-4}$, 10$^{-3}$, 10$^{-2}$, and 10$^{-1}$. Each "plateau" visible for certain solvers in Figure 1a represents a different relative tolerance setting, with tighter tolerances leading to higher SDAs."**

• All of the figures showing land area should have axis markers (latitutde and longitude, if possible, or at least the X and Y dimension indices). Once these are in place, the grid index given on page 11, line 5 should be changed to match the dimensions used for the figures, so that the reader can identify which point on the grid is being discussed.

**Grid cell coordinates have been added to figures 3, 4, 6, and 10. Cell coordinates for plots in figure 5 are now referenced on page 12, lines 5-6 (p 11, line 4 in original document)**

• Table 2: There are some typesetting issues here, particularly where a subscript should or should not have been used. Note, for example, the "Other information" entry for wet deposition.

**In Table 2, under Wet deposition "Other information" the unit "(s)" is no longer a subscript**

References
Djouad, R. and Michelangeli, D. V.: Investigation of splitting gas and aqueous operators in atmospheric multiphase box models, Atmos. Res., 71(4), 253–263, 2004.

---

## Author Comment (AC3) · 2 Mar 2017

"A framework for expanding aqueous chemistry in the Community Multiscale Air Quality (CMAQ) model version 5.1" by Kathleen M. Fahey et al.

Response to reviewer #3

**The authors thank reviewer #3 for their comments and support for publication. In the following, we restate the reviewer comments and follow with our response in "bold" font.**

Fahey et al. presents an implementation of two new cloud aqueous phase chemistry options (AQCHEM-KMT and AQCHEM-KMTI) for the CMAQ model. The paper is well written and will be of interest to the readers of GMD, I therefore support its publication in GMD after issues raised Referees #1 and #2 have been addressed.

**We have attempted to address all of the comments/suggestions from reviewers 1 and 2. One can find additional details in the posted responses to those reviewers.**

Minor issues: - Page 9, Line 29: Should "in-line calculation of photolysis rates" be "online calculations of photolysis rates"?

**To be consistent with descriptions of the CMAQ photolysis module published elsewhere (e.g., Appel et al., 2016), we refer to these as "in-line" calculations. We have updated the other references to "inline" processes in this section to "in-line" as well.**

**References:**

**Appel, K.W., Napelenok, S.L., Foley, K.M., Pye, H.O.T., Hogrefe, C., Luecken, D.J., Bash, J.O., Roselle, S.J., Pleim, J.E., Foroutan, H., Hutzell, W.T., Pouliot, G.A., Sarwar, G., Fahey, K.M., Gantt, B., Gilliam, R.C., Kang, D., Mathur, R., Schwede, D.B., Spero, T.L., Wong, D.C., and J.O. Young (2016) Overview and evaluation of the Community Multiscale Air Quality (CMAQ) model version 5.1, Geosci. Model Dev. Discuss., doi:10.5194/gmd-2016-226.**

---

## Author Comment (AC2)

"A framework for expanding aqueous chemistry in the Community Multiscale Air Quality (CMAQ) model version 5.1" by Kathleen M. Fahey et al.

Response to reviewer #2

**The authors thank reviewer #2 for their comments and suggested revisions.  In the following, we restate the reviewer comments and follow with our response in "bold" font.  Note that any references to page/line number are made for both the "updated" manuscript (with markup) and the original discussion paper.**

Fahey et al. developed a new cloud chemistry mechanism (AQCHEM-KMT) for use in large-scale models and implemented it into CMAQ. AQCHEM-KMT allows for the investigation of aqueous-phase chemical reactions that could not be implemented into the aqueous-phase chemistry mechanism that is currently implemented in CMAQ (AQCHEM) because of its explicit consideration of processes such as mass transfer limitations. As most large-scale models use an aqueous-phase chemical mechanism similar to AQCHEM, this development represents a significant step forward in examining the influence of aqueous-phase chemical reactions on the chemical composition of the atmosphere in large-scale models. The paper is well written, and I recommend publication in GMD after some very minor issues are addressed.

The authors find much larger differences between AQCHEM-KMT and AQCHEM for sulfate than for SOA, but the reasons for this difference could be more explicitly discussed than they currently are. What other aqueous-phase reactions or types of reactions can now be implemented in AQCHEM-KMT that was not possible with AQCHEM?

**AQCHEM-KMT would be well-suited to represent additional S(IV) oxidation chemistry as well as other species' chemistry (like oxidant or nitrogen chemistry) not currently represented in CMAQ cloud chemistry at all.  A future application for AQCHEM-KMT will likely be to replace the "parameterized" in-cloud SOA production from glyoxal and methylglyoxal with an explicit/mechanistic representation of that chemistry. While in-cloud production of organic acids that remain in the aerosol phase after cloud droplet evaporation has been widely studied in laboratory and modeling experiments (e.g., Lim et al., 2010), implementation of an explicit representation of such a chemical mechanism is something that could not easily be done with AQCHEM and its forward Euler solver, in part due to the increased stiffness of the system (and thus the implementation of the simple parameterization of in-cloud SOA chemistry in AQCHEM in CMAQv4.7).**

**This parameterization typically predicts only low surface level concentrations of in-cloud SOA on average and may be more "episodic" in nature than $SO_4^{2-}$.  While $SO_4^{2-}$ production in-cloud is a major production pathway for atmospheric $SO_4^{2-}$ (potentially dominating over gas phase production) and contributes significantly to predicted average surface $SO_4^{2-}$ concentrations, in-cloud SOA will likely only be a small fraction of total SOA production on average, in part because of the "transient" nature of clouds as well as the fact that the only chemical pathways represented in the model for in-cloud SOA production are photochemically driven (and thus may only be important during only a fraction of the day or only certain seasons (e.g., summer)).  As mentioned in section 3.1, in the parameterization for in-cloud SOA formation, aqueous phase OH concentrations are held constant during cloud processing, so there is a reduced sensitivity to mass transfer limitations for those reactions, because they do not consider mass transfer limitations for OH.**

**We have added some additional text to section 3.1 (new text is underlined)**

**Pate 12, line 19: "Absolute ORGC mass predictions are less impacted than $SO_4^{2-}$, but these tend to be low on average in the base case and may have limited sensitivity to changes in mass transfer treatment due in part to CMAQ's implementation of cloud SOA formation…"**

**Page 12, lines 22-29 (p 11, line 18 in original document): "The hydroxyl radical concentration is estimated at the start of cloud processing based on the initial gas phase concentration (Henry's law) and held constant for the duration of the "master" cloud time step (i.e., mass transfer limitations are not considered for OH).  This was done in part to compensate for the lack of a more complete treatment of radical/organic chemistry in the aqueous phase, along with a relatively loose coupling between gas and aqueous chemistry in CMAQ.  A constant oxidant concentration may cause an artificially high rate of consumption of the precursor species**

**and insensitivity of the reaction to droplet size and associated mass transfer limitations. In fact, it has been suggested that in-cloud oxidation of organic species by OH may be oxidant limited due in part to the effects of mass transfer limitations on aqueous OH concentrations (Ervens et al., 2014)…"**

Are there other sulfate production mechanisms that are better suited for AQCHEM-KMT than AQCHEM that are not currently included in the model, or do the authors expect sulfate to almost always be similar for AQCHEM-KMT and AQCHEM on monthly time scales?

**There are additional aqueous phase production pathways for $SO_4^{2-}$ that are not included in AQCHEM that may be an important contributor to $SO_4^{2-}$ concentrations in certain environments, such as the oxidation of $SO_2$ by $HNO_4$ (Leriche et al., 2003) or $NO_2$ or aqueous oxidation of $SO_2$ by OH (Seinfeld and Pandis, 2006). These may have an impact on the monthly average $SO_4^{2-}$ values but would need to be implemented and investigated further to determine whether they either lead to a regional average increase or if they lead to more episodic changes and have limited impacts on average $SO_4^{2-}$ predictions.**

In the introduction, it would be good to specify that when you refer to the aqueous phase you are referring specifically to cloud droplets and not liquid water associated with aerosols

**Page 2, line 8-9: Changed "aqueous phase production of $SO_4^{2-}$ dominates" to "aqueous phase production of $SO_4^{2-}$ in cloud and fog droplets dominates"**

**Page 2, lines 10 and 11: Changed "aqueous" to "in-cloud"**

**Page 2, Lines 13-14: Changed "potentially significant role that aqueous pathways may have on the formation" to "potentially significant role that aqueous pathways (in cloud droplets and wet aerosols) may have on the formation"**

**Page 3, line 31 (p 3, line 28 in original document): Changed "mass transfer between the gas and aqueous phases" to "mass transfer between the gas phase and cloud droplets"**

**Page 3, line 34 (p 3, lines 30-31 in original document): Changed "these additional aqueous-phase chemistry options" to "these additional in-cloud aqueous-phase chemistry options"**

**Page 4, line 3 (p 3, line 33 in original document): Changed "updated aqueous chemistry options" to "updated cloud chemistry options"**

**Page 4, line 9 (p 4, line 6 in original document): Specified that we are applying KPP to the "in-cloud" aqueous phase chemical mechanism**
.
Page 3 line 13: Refer to Table S3 so one can readily find the seven oxidation reactions, or list them here.

**We have added the following sentence after referring to the seven oxidation reactions on page 3: "The seven reactions represented are the oxidation of aqueous $SO_2$ by hydrogen peroxide, ozone, oxygen (catalyzed by iron and manganese), methylhydroxyperoxide, and peroxyacetic acid as well as two reactions that parameterize SOA formation from glyoxal and methylglyoxal."**

Many abbreviations/acronyms in the text and tables are not defined (e.g., MPAN). Perhaps add a table of abbreviations/acronyms in the SI.

**We have attempted to find all the abbreviations/acronyms in the text and define them (at least) at their first mention. These include the following updates:**

**On page 1, line 30: Changed "during the SOAS field campaign period" to "during the Southern Oxidant and Aerosol Study (SOAS) period"**

On page 2, line 18 (p 2, line 17 in original document): Changed "more than half of the total $PM_{2.5}$ concentration" to "more than half of the total fine particulate matter ($PM_{2.5}$) concentration"

On page 3, line 14: Changed S(VI) to "sulfate"

On page 3, line 18 (p 3, line 16 in original document):  Changed "stiff systems of ODEs" to "stiff systems of ordinary differential equations (ODEs)"

On page 4 line 21 (p 4, line 18  in original document): Changed S(VI) to $SO_4^{2-}$

On page 6, line 8 (p 6, line 4 in original document): Changed "i.e., IEPOX/MAE" to "i.e., isoprene epoxydiol/methacrylic acid epoxide chemistry"

On page 6, line 22 (p 6, line 18 in original document): Changed "as a successor to LSODE" to "as a successor to the Livermore Solver for Ordinary Differential Equations (LSODE)"

On page 11, line 1 (p 10, line 4 in original document):  Changed "formation from IEPOX and MPAN products" to "formation from IEPOX and methacryloylperoxynitrate (MPAN) products"

On page 11, line 11 (p 10, line 12 in original document): changed "2013-specific EGU continuous…" to "2013-specific electric generating unit (EGU) continuous…"

On page 11, line 12 (p 10, line 13 in original document): changed "with BELD4 land cover" to "with Biogenic Emissions Land Use Database (BELD4) land cover"

On page 14, lines 7-8 (p 13, line 1 in original document): changed "$\Delta SOA_{IEPOX/MPAN}$, $\Delta SOA_{2-MG}$" to "change in total SOA from the IEPOX/MPAN pathways ($\Delta SOA_{IEPOX/MPAN}$), change in 2-MG ($\Delta SOA_{2-MG}$)"

References:

Leriche, M., Deguillaume, L., and N. Chaumerliac (2003) Modeling study of strong acids formation and partitioning in a polluted cloud during wintertime.  J. Geophys. Res., 108(D14), doi:10.1029/2002JD002950

Lim, Y.B., Tan, Y., Perri, M.J., Seitzinger, S.P., and B.J. Turpin (2010) Aqueous chemistry and its role in secondary organic aerosol (SOA) formation. Atmos. Chem. Phys., 10, 10521-10539.

Seinfeld, J. and S. N. Pandis (2006) Atmospheric Chemistry and Physics: From Air Pollution to Climate Change, 2nd ed. John Wiley and Sons, Inc.